# Graph Classification Gaussian Processes via Hodgelet Spectral Features

**Mathieu Alain**[1,5,†]    **So Takao**[2]    **Bastian Rieck**[3]    **Xiaowen Dong**[4]    **Emmanuel Noutahi**[5]

[1]University College London    [2]California Institute of Technology    [3]University of Fribourg
[4]University of Oxford    [5]Valence Labs
[†]mathieu.alain.21@ucl.ac.uk

## Abstract

The problem of classifying graphs is ubiquitous in machine learning. While it is standard to apply graph neural networks or graph kernel methods, Gaussian processes can be employed by transforming spatial features from the graph domain into spectral features in the Euclidean domain, and using them as the input points of classical kernels. However, this approach currently only takes into account features on vertices, whereas some graph datasets also support features on edges. In this work, we present a Gaussian process-based classification algorithm that can leverage one or both vertex and edges features. Furthermore, we take advantage of the Hodge decomposition to better capture the intricate richness of vertex and edge features, which can be beneficial on diverse tasks.

## 1 Introduction

Classification is omnipresent in machine learning, yet typically assumes data to be Euclidean. Extending this task to non-Euclidean domains, such as graphs, presents challenges due to their irregularity, varying sizes, and multi-site information (e.g. vertices and edges). However, classifying graphs is of critical importance in scientific and industrial applications, being used for instance, to predict properties of molecules, or discovering new drugs [1, 2]. Although graph neural networks [3] are usually the model of choice for such applications, a downside is that they may require large datasets for effective training. Gaussian processes (GP) [4], on the other hand, prove to be a data-efficient and interpretable modelling choice. They do not need separate validation datasets to tune hyperparameters and provide robust uncertainty estimates for predictions. This makes them ideal for small-data scenarios and high-risk decision-making tasks that require reliable uncertainty estimates.

In a recent work, Opolka et al. [5] introduced a GP-based algorithm capable of classifying graphs. Their method relies on tools developed from the graph signal processing literature [6], including the spectral graph Fourier transform [7] and the spectral graph wavelet transform [8]. Specifically, spectral graph methods use spatial graph features to compute graph spectral coefficients, which may be leveraged to generate spectral features in the Euclidean domain. Such spectral representations can be passed as input points to a standard GP [5], subsequently employed for classification via approximate inference [9]. Closely related are techniques developed in the early graph neural network literature, for instance, Spectral Networks [10] and ChebNet [11], which lean on the graph Fourier transform and graph wavelet transform to establish a notion of convolution on graphs. While the approach proposed in Opolka et al. [5] accommodates features living on vertices, it cannot easily take into account features on edges. However, edge information can often be as valuable as vertex information, representing crucial quantities such as flows [12, 13] and chemical bonds [14].

Our paper aims to fill this gap by proposing a novel GP-based classification algorithm that naturally incorporate features on vertices, edges, and more generally, simplices, building on recent work defining GPs on simplicial complexes [15] and cellular complexes [16]. Moreover, we utilise the

Workshop on Bayesian Decision-making and Uncertainty, 38th Conference on Neural Information Processing Systems (NeurIPS 2024).

celebrated Hodge decomposition on spatial graph features, separating and processing them into three canonical components, each exhibiting distinct properties. This enables the modelling of these components separately, using different kernels. This idea have been used successfully in past works [17–20, 15], providing greater flexibility. We are not aware that these recent techniques have been applied in the context of graph classification. We demonstrate empirically that these extensions achieve similar or better performance than the method introduced by Opolka et al. [5]. Although we focus on graphs, our approach can be extended easily to higher-order networks, such as simplicial complexes, cellular complexes, and hypergraphs, which can describe polyadic interactions [21–24], thereby generalising the dyadic interactions found in graphs (see Appendix D). We highlight that our method can be readily adapted for regression by selecting an appropriate likelihood. Finally, although our Hodgelet spectral features are employed in the context of Gaussian processes, they can be readily integrated into other machine learning methods.

## 2 Gaussian Processes for Graph Classification

We assume a dataset containing $M$ undirected graphs $\mathcal{G}^{(1)}, \ldots, \mathcal{G}^{(M)}$ and labels $y^{(i)} \in \mathbb{Z}$, for $1 \leqslant i \leqslant M$. We assign an orientation to each graph but emphasise that this choice is arbitrary. Let $\mathcal{V}^{(i)}$ be the vertex set and $\mathcal{E}^{(i)}$ be the edge set, both finite. For each graph $\mathcal{G}^{(i)} = (\mathcal{V}^{(i)}, \mathcal{E}^{(i)})$, there are $N_v^{(i)}$ vertices, $N_e^{(i)}$ edges, and an incidence matrix $\boldsymbol{B}_{ve}^{(i)} \in \mathbb{Z}^{N_v^{(i)} \times N_e^{(i)}}$. The latter encodes the incidence between each vertex and edge, and furthermore defines the *graph Laplacian* $\boldsymbol{L}_v^{(i)} \coloneqq \boldsymbol{B}_{ve}^{(i)} \boldsymbol{B}_{ve}^{(i)\top} \in \mathbb{Z}^{N_v^{(i)} \times N_v^{(i)}}$. By considering 3-cliques, we obtain the edge-to-triangle incidence matrix $\boldsymbol{B}_{et}^{(i)} \in \mathbb{Z}^{N_e^{(i)} \times N_t^{(i)}}$, where $N_t^{(i)}$ is the number of triangles in $\mathcal{G}^{(i)}$. Likewise, we define the *graph Helmholtzian* $\boldsymbol{L}_e^{(i)} \coloneqq \boldsymbol{B}_{ve}^{(i)\top} \boldsymbol{B}_{ve}^{(i)} + \boldsymbol{B}_{et}^{(i)} \boldsymbol{B}_{et}^{(i)\top} \in \mathbb{Z}^{N_e^{(i)} \times N_e^{(i)}}$, which applies to edges rather than vertices. We underline that $\boldsymbol{L}_v^{(i)}$ and $\boldsymbol{L}_e^{(i)}$ are instances of the *discrete Hodge Laplacian* (see Appendix D.2). Let $\mathcal{V}^{(i)} \to \mathbb{R}^{D_v}$ and $\mathcal{E}^{(i)} \to \mathbb{R}^{D_e}$ be two functions, for $D_v, D_e \in \mathbb{N}$. By introducing an ordering on vertices and edges, we represent the preceding functions as matrices $\mathbb{R}^{D_v \times N_v^{(i)}}$ and $\mathbb{R}^{D_e \times N_e^{(i)}}$, respectively, which are understood as vertex and edge feature matrices, containing $D_v$ and $D_e$ *channels*, respectively. We denote vertex features for channel $1 \leqslant d \leqslant D_v$ by the column vector $\boldsymbol{x}_{vd}^{(i)} \in \mathbb{R}^{N_v^{(i)}}$ and edge features for channel $1 \leqslant d \leqslant D_e$ by the column vector $\boldsymbol{x}_{ed}^{(i)} \in \mathbb{R}^{N_e^{(i)}}$. We stress that our approach can adapt to graphs that may have vertex features, edge features, or both.

### 2.1 Wavelet transforms on graphs

The key idea behind our graph classification algorithm is to convert vertex and edge features from the graph domain into *spectral features* in the Euclidean domain, enabling standard GP classification. We first consider the eigendecomposition of the graph Laplacian and graph Helmholtzian

$$\boldsymbol{L}_v^{(i)} = \boldsymbol{U}_v^{(i)} \boldsymbol{\Lambda}_v^{(i)\top} \boldsymbol{U}_v^{(i)\top}, \qquad \boldsymbol{L}_e^{(i)} = \boldsymbol{U}_e^{(i)} \boldsymbol{\Lambda}_e^{(i)} \boldsymbol{U}_e^{(i)\top}, \tag{1}$$

and define *graph Fourier coefficients* as projections of spatial graph features onto the eigenbases,

$$\hat{\boldsymbol{x}}_{vd}^{(i)} \coloneqq \boldsymbol{U}_v^{(i)\top} \boldsymbol{x}_{vd}^{(i)} \in \mathbb{R}^{N_v^{(i)}}, \qquad \hat{\boldsymbol{x}}_{ed}^{(i)} \coloneqq \boldsymbol{U}_e^{(i)\top} \boldsymbol{x}_{ed}^{(i)} \in \mathbb{R}^{N_e^{(i)}}. \tag{2}$$

We observe that $\hat{\boldsymbol{x}}_{\bullet d}^{(i)}$ reside in the eigenspace of $\boldsymbol{L}_\bullet^{(i)}$. Furthermore, they are *invariant* to vertex and edge ordering, making them sound choices for constructing spectral features, and they are perfectly localised in frequency, capturing *global* properties of the original features. However, it is often beneficial to also possess spatially localised information, focusing on *local* properties. A solution is to compute the more flexible *graph wavelet coefficients* [8] (see Appendix B) by modulating Fourier coefficients using a *wavelet filter* on the eigenvalues $\boldsymbol{\Lambda}_v^{(i)}$ and $\boldsymbol{\Lambda}_e^{(i)}$, and then perform the inverse *Fourier transform*. A wavelet filter is a combination of a *scaling function* at a single scale and a *wavelet function* at multiple scales, offering *multi-scale resolution*. Wavelet functions, $b : \mathbb{R} \to \mathbb{R}$ and $d : \mathbb{R} \to \mathbb{R}$, operate as *band-pass filters*, and scaling functions, $a : \mathbb{R} \to \mathbb{R}$ and $c : \mathbb{R} \to \mathbb{R}$, are *low-pass filters*. A wavelet filter captures one perspective of a graph, but to obtain a comprehensive

picture, it is essential to employ multiple wavelet filters. Wavelet filters $j$ are defined by

$$w_{vj}(\lambda) := a(\alpha_j \lambda) + \sum_{l=1}^{L_v} b(\beta_{jl}\lambda), \quad \Theta_{vj} := \{\alpha_j, \beta_{j1}, \ldots, \beta_{jL_v}\}, \quad 1 \leqslant j \leqslant W_v, \tag{3}$$

$$w_{ej}(\lambda) := c(\gamma_j \lambda) + \sum_{l=1}^{L_e} d(\delta_{jl}\lambda), \quad \Theta_{ej} := \{\gamma_j, \delta_{j1}, \ldots, \delta_{jL_e}\}, \quad 1 \leqslant j \leqslant W_e, \tag{4}$$

where $L_\bullet$ is the number of scales, $W_\bullet$ is the number of wavelet filters, and $\Theta_{\bullet j}$ is a collection of trainable parameters controlling the scaling. Wavelet coefficients are given by

$$\hat{\boldsymbol{x}}_{vdj}^{(i)} := \boldsymbol{U}_v^{(i)} w_{vj}\big(\boldsymbol{\Lambda}_v^{(i)}\big)\boldsymbol{U}_v^{(i)\top}\boldsymbol{x}_{vd}^{(i)} \in \mathbb{R}^{N_v^{(i)}}, \qquad \hat{\boldsymbol{x}}_{edj}^{(i)} := \boldsymbol{U}_e^{(i)} w_{ej}\big(\boldsymbol{\Lambda}_e^{(i)}\big)\boldsymbol{U}_e^{(i)\top}\boldsymbol{x}_{ed}^{(i)} \in \mathbb{R}^{N_e^{(i)}}. \tag{5}$$

## 2.2 Hodge decomposition

The *discrete Hodge decomposition* [25] (see Appendix D.3) states that *spatial graph feature spaces*, i.e. the spaces inhabited by $\boldsymbol{x}_{vd}^{(i)}$ and $\boldsymbol{x}_{ed}^{(i)}$, can each be separated into an orthogonal sum of three subspaces, *exact*, *co-exact*, and *harmonic*, collectively referred to as the *Hodge subspaces*. From this, the eigenbases in (1) can be divided into sub-eigenbases, each spanning a different Hodge subspace,

$$\boldsymbol{U}_v^{(i)} = \big[\boldsymbol{U}_{vc}^{(i)}\ \boldsymbol{U}_{vh}^{(i)}\big], \qquad \boldsymbol{U}_e^{(i)} = \big[\boldsymbol{U}_{ee}^{(i)}\ \boldsymbol{U}_{ec}^{(i)}\ \boldsymbol{U}_{eh}^{(i)}\big]. \tag{6}$$

We observe that $\boldsymbol{U}_v^{(i)}$ has only two components. The *co-exact sub-eigenbasis* $\boldsymbol{U}_{vc}^{(i)}$ amounts to the non-zero eigenvectors of $\boldsymbol{L}_v^{(i)}$, and the *harmonic sub-eigenbasis* $\boldsymbol{U}_{vh}^{(i)}$ to the zero ones. The *exact* and co-exact *sub-eigenbases* $\boldsymbol{U}_{ee}^{(i)}$ and $\boldsymbol{U}_{ec}^{(i)}$ are the non-zero eigenvectors of $\boldsymbol{B}_{ve}^{(i)\top}\boldsymbol{B}_{ve}^{(i)}$ and $\boldsymbol{B}_{et}^{(i)}\boldsymbol{B}_{et}^{(i)\top}$, respectively. Finally, the harmonic sub-eigenbasis $\boldsymbol{U}_{eh}^{(i)}$ comprises the zero eigenvectors of $\boldsymbol{L}_e^{(i)}$. For edges, the exact and co-exact components are sometimes termed *gradient* and *curl* components, respectively, reminiscent of vector fields. A gradient part is *curl-free*, indicating no *vortices*. A curl part is *divergence-free*, meaning no *sources* or *sinks*. A harmonic part is curl-free and divergence-free.

## 2.3 Hodgelet spectral features

We generate graph spectral features from graph wavelet coefficients and then use them in downstream GP classification tasks (see Appendix A). By combining the wavelet transform (5) and the Hodge decomposition (6), we compute the wavelet coefficients $\hat{\boldsymbol{x}}_{vdjc}^{(i)}, \hat{\boldsymbol{x}}_{vdjh}^{(i)}$ and $\hat{\boldsymbol{x}}_{edje}^{(i)}, \hat{\boldsymbol{x}}_{edjc}^{(i)}, \hat{\boldsymbol{x}}_{edjh}^{(i)}$ (see Appendix C for more details). We derive our *Hodgelet spectral features* by concatenating the 2-norm of the preceding wavelet coefficients across each wavelet filter and channel, resulting in the column vectors $\boldsymbol{v}_c^{(i)}, \boldsymbol{v}_h^{(i)} \in \mathbb{R}^{W_v D_v}$ and $\boldsymbol{e}_e^{(i)}, \boldsymbol{e}_c^{(i)}, \boldsymbol{e}_h^{(i)} \in \mathbb{R}^{W_e D_e}$. These spectral representations, which are invariant to graph isomorphism, are then fed to our additive *Hodgelet kernel*

$$\begin{aligned}
\kappa\big(\mathcal{G}^{(i)}, \mathcal{G}^{(j)}\big) := {}&\kappa_{vc}\big(\boldsymbol{v}_c^{(i)}, \boldsymbol{v}_c^{(j)}\big) + \kappa_{vh}\big(\boldsymbol{v}_h^{(i)}, \boldsymbol{v}_h^{(j)}\big) \\
&+ \kappa_{ee}\big(\boldsymbol{e}_e^{(i)}, \boldsymbol{e}_e^{(j)}\big) + \kappa_{ec}\big(\boldsymbol{e}_c^{(i)}, \boldsymbol{e}_c^{(j)}\big) + \kappa_{eh}\big(\boldsymbol{e}_h^{(i)}, \boldsymbol{e}_h^{(j)}\big),
\end{aligned} \tag{7}$$

where $\kappa_{\bullet\bullet}$ is a standard kernel function, such as the *squared exponential kernel function*. We note that parameters $\Theta_{\bullet j}$ are optimised jointly with the kernel hyperparameters and that a separate kernel for each part of the Hodge decomposition offers greater flexibility. We highlight that our GP-based classification algorithm supports multi-dimensional spatial graph features and graphs of varying sizes, in contrast to typical graph kernel-based methods [26]. The GP component scales according to the number of graphs, while the eigendecompositions are a one-off cost that can be performed in advance.

## 3 Experiments

The aim of the experiments is two-fold: (1) we validate the added flexibility given by the Hodge decomposition, and (2) we demonstrate that when edge features are present, it is better to work with edges directly rather than converting graphs to line-graphs. The latter point has been observed in previous works [22, 15] and our experiments, across 10 seeds, further validate their conclusions.

| | ENZYMES | MUTAG | IMDB-BINARY | IMDB-MULTI | ring-vs-clique | sbm |
|---|---|---|---|---|---|---|
| WT-GP | 65.00 ± 4.94 | 86.73 ± 4.18 | **74.20 ± 3.87** | 48.73 ± 2.76 | 99.5 ± 1.5 | 86.42 ± 6.92 |
| WT-GP-Hodge | **67.65 ± 6.86** | **88.06 ± 7.99** | 73.40 ± 3.04 | **52.09 ± 3.44** | **100.0 ± 0.0** | **88.02 ± 7.35** |

Table 1: Comparison of classification accuracy on several graph classification benchmark datasets.

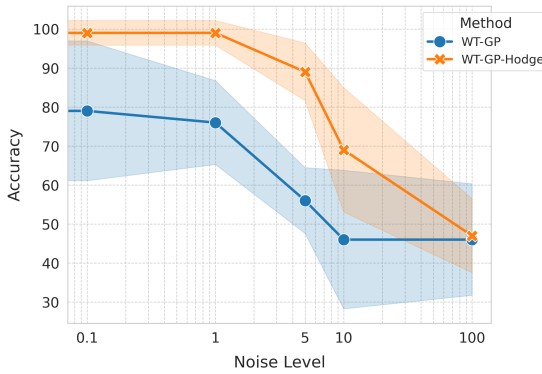

| $N$ | WT-GP-LG | WT-GP-Hodge |
|---|---|---|
| 25 | 51.0 ± 16.4 | 50.0 ± 14.1 |
| 50 | 53.0 ± 17.9 | 73.0 ± 12.7 |
| 75 | 56.0 ± 14.3 | 85.0 ± 12.8 |
| 100 | 54.0 ± 15.0 | 89.0 ± 10.4 |
| 150 | 57.0 ± 13.5 | 92.0 ± 7.0 |
| 200 | 54.0 ± 12.0 | 94.0 ± 8.0 |
| 250 | 58.0 ± 16.6 | 95.0 ± 7.2 |
| 300 | 56.0 ± 12.8 | 93.5 ± 7.8 |
| 350 | 53.0 ± 13.3 | 94.8 ± 7.4 |

Figure 1: Accuracy vs. noise level in the vector field classification.

Table 2: Accuracy of vector field classification.

## 3.1 Graph classification benchmarks

Our first experiment compares the method in Opolka et al. [5], which we refer to the wavelet-transform GP (WT-GP), which does not use the Hodge decomposition, to our method, WT-GP-Hodge, which employs the decomposition. In Table 1, we display the results on some standard graph classification benchmark datasets used in Opolka et al. [5]. We observe that on all but one dataset, WT-GP-Hodge improves the classification accuracy. This may be surprising as the Hodge decomposition for vertex features yields only co-exact and harmonic parts, where the harmonic part is constant across the connected components of the graph. This suggests that we can gain accuracy by separating vertex features into a constant bias (harmonic part) and fluctuations around it (co-exact part), using different kernels for each. However, if constant biases do not aid classification (e.g. when classes have vertex feature of similar magnitudes), then we do not expect WT-GP-Hodge to improve over WT-GP.

## 3.2 Vector field classification

Our second experiment consider the task of classifying noisy vector fields, i.e. predicting whether they are predominantly divergence-free or curl-free. We proceed by generating 100 random vector fields, with half mostly divergence-free (Figure E.1c) and the other half mostly curl-free (Figure E.1d). The generated vector fields are then projected onto the edges of a randomly generated triangular mesh on a square domain with $N$ vertices (Figure E.2). Finally, we corrupt the edge features with i.i.d. Gaussian noise, resulting in a dataset composed of 100 oriented graphs, each containing scalar edge features corresponding to the *net flow* of the vector field along the edges. Again, we compare our method against the vanilla WT-GP classification method. However, since WT-GP does not take edge features, we first convert graphs to line-graphs before applying it. We refer to it as WT-GP-LG. In Table 2, we display the results of WT-GP-LG and WT-GP-Hodge for various choices of $N$. We see that WT-GP-Hodge is consistently better, with large improvements as mesh resolution is increased. On the other hand, WT-GP-LG cannot distinguish accurately between divergence-free and curl-free fields, even as the mesh resolution becomes higher. Likely reasons: (1) the Hodge decomposition in WT-GP-Hodge helps to discriminate more clearly between divergence-free and curl-free components, and (2) there are properties that are canonical to edges, such as orientation, which WT-GP-Hodge can handle naturally, whereas WT-GP-LG cannot. In Figure 1, we plot the classification accuracy with varying noise level, which shows robustness of WT-GP-Hodge to noise compared to WT-GP-LG.

## 4  Conclusion and Future Directions

We have presented a GP-based classification algorithm for classifying graphs according to one or both vertex and edge features. By applying the graph wavelet transform to spatial graph features, we have constructed spectral features, providing multi-resolution spectral signatures of the original features, subsequently utilised as input points to a standard GP. Furthermore, by taking the discrete Hodge decomposition, we have shown improvements over the method proposed by Opolka et al. [5], even on graph datasets containing only vertex features, owing to our flexible Hodgelet kernel. Overall, we have demonstrated that our approach effectively improves graph classification tasks by employing a spectral perspective to capture both local and global properties of vertex and edge features. In the future, we intend to explore extensions to higher-order networks, including simplicial complexes, cellular complexes, and hypergraphs, which we briefly outline in Appendix D.

## Acknowledgments

MA is supported by a Mathematical Sciences Doctoral Training Partnership held by Prof. Helen Wilson, funded by the Engineering and Physical Sciences Research Council (EPSRC), under Project Reference EP/W523835/1. ST is supported by a Department of Defense Vannevar Bush Faculty Fellowship held by Prof. Andrew Stuart, and by the SciAI Center, funded by the Office of Naval Research (ONR), under Grant Number N00014-23-1-2729.

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

# A Gaussian Processes

Gaussian processes are stochastic processes that can be considered as Gaussian distributions extended to *infinite dimensions*, providing a powerful *non-parametric* technique for modelling distributions over functions.

More precisely, a Gaussian process over $\mathbb{R}^d$ is a random function $f : \Omega \times \mathbb{R}^d \to \mathbb{R}$, characterised by a *mean function* $\mu : \mathbb{R}^d \to \mathbb{R}$ and a *kernel function* $\kappa : \mathbb{R}^d \times \mathbb{R}^d \to \mathbb{R}$, satisfying $\mu(\boldsymbol{x}) := \mathbb{E}[f(\boldsymbol{x})]$ and $\kappa(\boldsymbol{x}, \boldsymbol{x}') := \mathrm{Cov}[f(\boldsymbol{x}), f(\boldsymbol{x}')]$, respectively, for any $\boldsymbol{x}, \boldsymbol{x}' \in \mathbb{R}^d$. We denote this relation by writing that $f \sim \mathrm{GP}(\mu, \kappa)$. For any *finite* collection of *input points* $\boldsymbol{x}_1, \ldots, \boldsymbol{x}_M \in \mathbb{R}^d$, the random vector $\boldsymbol{f} = [f(\boldsymbol{x}_1), \ldots, f(\boldsymbol{x}_M)]^\top \sim \mathrm{N}(\boldsymbol{\mu}, \boldsymbol{K})$ is *jointly* Gaussian, where $\boldsymbol{\mu} = [\mu(x_1), \ldots, \mu(x_M)]^\top$ and $(\boldsymbol{K})_{ij} = k(\boldsymbol{x}_i, \boldsymbol{x}_j)$.

Gaussian processes are non-parametric in the sense that they define a distribution directly on functions rather than on parameters of a function. This offers the advantage that a Gaussian process can generate functions of increasing sophistication as more data are acquired, unconstrained by a predetermined parametric structure.

For a complete presentation of Gaussian processes, we refer the reader to the book by Rasmussen and Williams [4].

## A.1 Bayesian inference paradigm

Gaussian processes are employed in the *Bayesian inference paradigm*:

1. A Gaussian process is defined as a *prior distribution* over an *unknown function*, capturing our *prior beliefs* about the possible shapes and behaviour of this function before observing any data.

2. A *likelihood* is a modelling choice representing how the functions from the prior distribution generate the data. An essential role of the likelihood is to define the assumptions about the *data noise*. By observing a *dataset*, the likelihood assesses the goodness of fit between our prior beliefs and the actual *observed data*.

3. By combining the prior distribution and the likelihood using the celebrated *Bayes' theorem*, a *posterior distribution* is computed to *update* our beliefs about the unknown function. For regression, a common assumption is a Gaussian likelihood, leading the posterior distribution to be a Gaussian process again. However, for classification, the likelihood is non-Gaussian, often Bernoulli for *binary classification* or categorical for *multi-class classification*. This results in a posterior distribution that is *intractable*, meaning that it cannot be expressed in a closed-form. In this case, it is possible to derive an approximate posterior distribution using techniques such as *variational inference*.

4. By conditioning on the dataset, the posterior distribution is employed to derive the *predictive posterior distribution* for new input points. The mean of this distribution serves as the predictions, while the variance provides a measure of the uncertainty around them.

## A.2 Kernel functions

Gaussian process is uniquely defined by a mean function and kernel function, which has the advantage of being simple and *interpretable*. The mean function is often set to zero, i.e. $\mu(\boldsymbol{x}) = 0$, for every $\boldsymbol{x} \in \mathbb{R}^d$, especially after normalising the dataset since it simplifies derivations and proofs. The kernel function is thus considered the most defining component in describing the behaviour of a Gaussian process and measures the similarities between pairs of input points. It is important to note that a kernel function must be a *symmetric* and *positive semi-definite* function as it defines the covariance between two input points. Some popular choices are the *square exponential kernel function* (also called *radial basis function kernel*)

$$\kappa_{\mathrm{se}}(\boldsymbol{x}, \boldsymbol{x}') := \sigma^2 \exp\left(-\frac{\|\boldsymbol{x} - \boldsymbol{x}'\|^2}{2\ell^2}\right), \quad \ell > 0, \tag{8}$$

and the *Matérn kernel function*

$$\kappa_{\mathrm{mat}}(\boldsymbol{x}, \boldsymbol{x}') := \sigma^2 \frac{2^{1-\nu}}{\Gamma(\nu)} \left(\sqrt{2\nu}\frac{\|\boldsymbol{x} - \boldsymbol{x}'\|}{\ell}\right)^\nu K_\nu\left(\sqrt{2\nu}\frac{\|\boldsymbol{x} - \boldsymbol{x}'\|}{\ell}\right), \quad \ell, \nu > 0, \tag{9}$$

where $\| \cdot \|$ is the 2-norm, $\Gamma$ the *gamma function*, and $K_\nu$ the *modified Bessel function* of the second kind. We observe that for $\nu \to \infty$, the Matérn kernel function converges to the square exponential kernel function.

The parameters of a kernel function are called *hyperparameters* and they are typically optimised by *maximum marginal likelihood estimation*.

### A.3   Computational cost and memory requirements

When the likelihood is Gaussian, a closed form expression of the posterior exists and computing it requires inverting a $N \times N$ *Gram matrix*, where $N$ is the number of training input points. This results to a cubic *computational complexity* $\mathcal{O}(N^3)$ and quadratic *memory requirements* $\mathcal{O}(N^2)$. Fortunately, sparse Gaussian processes [27] have been devised to alleviate this large computational expense by constructing a smaller training dataset consisting of $M$ *pseudo input points*, called *inducing points*, where $M \ll N$. This technique leads to a computational complexity $\mathcal{O}(NM^2 + M^3)$ and memory requirements $\mathcal{O}(MN)$. The sparse variational Gaussian process in [28] further extends this to non-Gaussian likelihood settings by means of stochastic variational inference (see also [29] for a variational approximation without using inducing points). This can be used to compute approximate posteriors in the setting of classification problems.

## B   Spectral Analysis

Spectral analysis is a set of tools for analysing and manipulating signals according to their constituent *frequencies*. These methods have found numerous applications in disciplines as diverse as telecommunications, physics, chemistry, signal processing, quantitative finance and, of course, machine learning.

Suppose a continuous function $f : \mathbb{R} \to \mathbb{R}$ representing a signal function. This signal function can be expressed as a superposition of *sinusoidal plane waves* $e^{i2\pi\xi x}$ at different frequencies $\xi$,

$$f(x) := \int_{-\infty}^{\infty} \hat{f}(\xi)e^{i2\pi\xi x}d\xi, \tag{10}$$

where $\hat{f}(\xi)$ is the *Fourier transform* of $f$,

$$\hat{f}(\xi) := \int_{-\infty}^{\infty} f(x)e^{-i2\pi\xi x}dx. \tag{11}$$

The absolute value $|\hat{f}(\xi)|$ represents the amplitude of $\xi$, reflecting the strength in the original signal of the sinusoidal plane wave associated to $\xi$. The transformed function $\hat{f}$ makes it possible to explore the importance of specific frequencies, revealing insights into patterns, periodicities and trends that may not be apparent from examining the original signal $f$. We observe that this *integral transform* does not lose information. Indeed, the function $f$ can be recovered from (11) using the *inverse Fourier transform* (10). The collection of sinusoidal plane waves $\{e^{i2\pi\xi x}\}_\xi$ are *orthonormal* and commonly called the *Fourier basis*.

**Fourier transform and Laplacian.**   There are important connections between the Fourier transform and the Laplacian. The Laplacian $\Delta f$ is the divergence of the gradient of $f$,

$$\Delta f := \nabla \cdot \nabla f. \tag{12}$$

It turns out that the Fourier basis function $e^{i2\pi\xi x}$ is the *generalised eigenfunction* of $\Delta f$ associated to the eigenvalue $\lambda = -(2\pi)^2|\xi|^2$. We note that small eigenvalues correspond to small frequencies, and large eigenvalues to large frequencies. Consequently, the eigenvalues of the Laplacian are good surrogates for the frequencies.

### B.1   Graph wavelet transform

The notion of Fourier transform can be adapted to graphs. Let $\boldsymbol{x} \in \mathbb{R}^N$ be a column vector representing a discrete signal on the $N$ vertices of a graph. The eigendecomposition of the *graph Laplacian* is given by

$$\boldsymbol{L} = \boldsymbol{U}\boldsymbol{\Lambda}\boldsymbol{U}^\top. \tag{13}$$

Similar to the continuous case, the eigenvector matrix $\boldsymbol{U}$ is referred to as the *graph Fourier basis*, and we consider the eigenvalues of $\boldsymbol{L}$ as the *graph frequencies*.

The *graph Fourier transform* $\boldsymbol{U}^T$ performs a projection of the signal onto the graph Fourier basis, thus producing a finite number of *Fourier coefficients*

$$\hat{\boldsymbol{x}} \coloneqq \boldsymbol{U}^\top \boldsymbol{x} \in \mathbb{R}^N. \tag{14}$$

We see that the Fourier coefficients belong to the eigenspace of the graph Laplacian. They provide *complete localisation in terms of frequency*, meaning that every frequency is represented and its contribution to the original signal can be identified, the $n$-th component of $\hat{\boldsymbol{x}}$ is associated to the $n$-th frequency. By contrast, the original signal offers *complete resolution in space* (or *time*), making it possible to determine how the signal varies across the vertices. In other words, the $n$-th component of $\boldsymbol{x}$ is associated to the $n$-th vertex. However, it is often advantageous not to be limited to just one or the other. The *spectral graph wavelet transform* offers *multi-scale resolution*, allowing a more balanced analysis between the spatial and frequency domains. Instead of sinusoidal plane waves, it employs the more general notion of *wavelets*. A wavelet can be scaled and translated, meaning that it can be tuned to capture localised changes in space. By contrast, a sinusoidal plane wave is uniform and extend across the entire signal.

The idea behind wavelet transform is to derive the *wavelet coefficients* from the Fourier coefficients by applying a *wavelet filter* $w : \mathbb{R} \to \mathbb{R}$ on the eigenvalues of the graph Laplacian and then performing the inverse Fourier transform,

$$\hat{\boldsymbol{x}} \coloneqq \boldsymbol{U} w(\boldsymbol{\Lambda}) \boldsymbol{U}^\top \boldsymbol{x}. \tag{15}$$

A wavelet filter is a combination of a *scaling function* $a : \mathbb{R} \to \mathbb{R}$ at the scale $\alpha$ and a *wavelet function* $b : \mathbb{R} \to \mathbb{R}$ at $L$ different scales $\beta_l$,

$$w(\lambda) \coloneqq a(\alpha\lambda) + \sum_{l=1}^{L} b(\beta_l \lambda), \qquad \Theta \coloneqq \{\alpha, \beta_1, \ldots, \beta_L\}. \tag{16}$$

The wavelet function at each scale represents a scaled and translated variant of a *mother wavelet*. The role of the wavelet function is to serve as a *band-pass filter*, covering medium and high frequencies. The scaling function is a *low-pass filter*, capturing low frequencies.

The reader is referred to Hammond et al. [8] for a more detailed description of the graph wavelet transform.

**Mother wavelet.** A popular choice of mother wavelets is the *Mexican hat wavelet*, also called *Ricker wavelet*. The Mexican hat wavelet is defined as the negative normalised second derivative of a Gaussian function,

$$b(\lambda) \coloneqq \frac{2}{\sqrt{3}\sigma \pi^{1/4}} \left(1 - \left(\frac{\lambda}{\sigma}\right)^2\right) e^{-\frac{\lambda^2}{2\sigma^2}}, \quad \sigma > 0. \tag{17}$$

## C   Hodgelet spectral features

Here, we provide further details about the Hodgelet spectral features introduced in Section 2.3. By the Hodge decomposition from Section 2.2 (more details in Appendix D.3), the wavelet coefficients in (5) can be decomposed into exact, co-exact and harmonic terms (note that vertex features have no exact component),

$$\hat{\boldsymbol{x}}_{vdj}^{(i)} = \left[\hat{\boldsymbol{x}}_{vdjc}^{(i)}, \hat{\boldsymbol{x}}_{vdjh}^{(i)}\right]^\top, \quad \hat{\boldsymbol{x}}_{edj}^{(i)} = \left[\hat{\boldsymbol{x}}_{edje}^{(i)}, \hat{\boldsymbol{x}}_{edjc}^{(i)}, \hat{\boldsymbol{x}}_{edjh}^{(i)}\right]^\top, \tag{18}$$

where $e, c, h$ in the last entry of the subscripts indicate exact, co-exact and harmonic components, respectively. We also recall that the indices $d, j$ correspond to the channel and filter dimensions, respectively. Concretely, the components are computed as

$$\hat{\boldsymbol{x}}_{vdj\bullet}^{(i)} = \boldsymbol{U}_{v\bullet}^{(i)} w_{vj\bullet}\big(\boldsymbol{\Lambda}_{v\bullet}^{(i)}\big) \boldsymbol{U}_{v\bullet}^{(i)\top} \boldsymbol{x}_{vd}^{(i)}, \quad \hat{\boldsymbol{x}}_{edj\bullet}^{(i)} = \boldsymbol{U}_{e\bullet}^{(i)} w_{ej\bullet}\big(\boldsymbol{\Lambda}_{e\bullet}^{(i)}\big) \boldsymbol{U}_{e\bullet}^{(i)\top} \boldsymbol{x}_{ed}^{(i)}, \tag{19}$$

which follow from the orthogonality of the decomposition.

Next, we compute the Hodgelet spectral features $\boldsymbol{v}_\bullet^{(i)} \in \mathbb{R}^{D_v W_v}$, $\boldsymbol{e}_\bullet^{(i)} \in \mathbb{R}^{D_e W_e}$, which are defined by a concatenation of the 2-norms, i.e. $\|\boldsymbol{x}\|_2^2 := x_1^2 + \ldots + x_N^2$,

$$\left(\boldsymbol{v}_\bullet^{(i)}\right)_{dj} = \|\hat{\boldsymbol{x}}_{vdj\bullet}^{(i)}\|_2, \quad \left(\boldsymbol{e}_\bullet^{(i)}\right)_{dj} = \|\hat{\boldsymbol{x}}_{edj\bullet}^{(i)}\|_2, \tag{20}$$

where the column vectors $\boldsymbol{v}_\bullet^{(i)}, \boldsymbol{e}_\bullet^{(i)}$ are indexed by $(d,j) \in \{1, \ldots, D_\bullet\} \times \{1, \ldots, W_\bullet\}$. We note that the features (20) are invariant under graph isomorphism, due to the following argument.

**Permutation invariance.** Let us consider the vertex features for a graph $\mathcal{G}^{(i)}$ and consider a graph isomorphism $\varphi : \mathcal{V}^{(i)} \to \mathcal{V}^{(i)'}$, where $\mathcal{G}^{(i)'} = (\mathcal{V}^{(i)'}, \mathcal{E}^{(i)'}) \cong \mathcal{G}^{(i)}$. In vector representation (i.e., considering $\mathcal{V}^{(i)} \cong \mathbb{R}^{N_v^{(i)}}$ and $\mathcal{V}^{(i)'} \cong \mathbb{R}^{N_v^{(i)}}$ by introducing an ordering on the vertices), the isomorphism is defined via a permutation matrix $\boldsymbol{P}^{(i)} \in \mathbb{R}^{N_v^{(i)} \times N_v^{(i)}}$, which is orthogonal, that is $\boldsymbol{P}^{(i)\top} \boldsymbol{P}^{(i)} = \boldsymbol{I}$, and therefore norm-preserving. According to this permutation, one can check the following transformations,

$$\boldsymbol{x}_{vd}^{(i)} \overset{\varphi}{\mapsto} \boldsymbol{P}^{(i)} \boldsymbol{x}_{vd}^{(i)}, \tag{21}$$

$$\boldsymbol{U}_{v\bullet}^{(i)} \overset{\varphi}{\mapsto} \boldsymbol{P}^{(i)} \boldsymbol{U}_{v\bullet}^{(i)} \boldsymbol{P}^{(i)\top}, \tag{22}$$

$$\boldsymbol{\Lambda}_{v\bullet}^{(i)} \overset{\varphi}{\mapsto} \boldsymbol{P}^{(i)} \boldsymbol{\Lambda}_{v\bullet}^{(i)} \boldsymbol{P}^{(i)\top}. \tag{23}$$

The last line also implies that $w_{vj\bullet}(\boldsymbol{\Lambda}_{v\bullet}^{(i)}) \overset{\varphi}{\mapsto} \boldsymbol{P}^{(i)} w_{vj\bullet}(\boldsymbol{\Lambda}_{v\bullet}^{(i)}) \boldsymbol{P}^{(i)\top}$, since $w_{vj\bullet}$ is given as a component-wise function. Then, we get

$$\left(\boldsymbol{v}_\bullet^{(i)}\right)_{dj} = \|\hat{\boldsymbol{x}}_{vdj\bullet}^{(i)}\|_2 = \|\boldsymbol{U}_{v\bullet}^{(i)} w_{vj\bullet}(\boldsymbol{\Lambda}_{v\bullet}^{(i)}) \boldsymbol{U}_{v\bullet}^{(i)\top} \boldsymbol{x}_{vd}^{(i)}\|_2 \tag{24}$$

$$\overset{\varphi}{\mapsto} \|(\boldsymbol{P}^{(i)} \boldsymbol{U}_{v\bullet}^{(i)} \boldsymbol{P}^{(i)\top})(\boldsymbol{P}^{(i)} w_{vj\bullet}(\boldsymbol{\Lambda}_{v\bullet}^{(i)}) \boldsymbol{P}^{(i)\top})(\boldsymbol{P}^{(i)} \boldsymbol{U}_{v\bullet}^{(i)} \boldsymbol{P}^{(i)\top}) \boldsymbol{P}^{(i)} \boldsymbol{x}_{vd}^{(i)}\|_2 \tag{25}$$

$$= \|\boldsymbol{P}^{(i)} \boldsymbol{U}_{v\bullet}^{(i)} \underbrace{(\boldsymbol{P}^{(i)\top} \boldsymbol{P}^{(i)})}_{=\boldsymbol{I}} w_{vj\bullet}(\boldsymbol{\Lambda}_{v\bullet}^{(i)}) \underbrace{(\boldsymbol{P}^{(i)\top} \boldsymbol{P}^{(i)})}_{=\boldsymbol{I}} \boldsymbol{U}_{v\bullet}^{(i)} \underbrace{(\boldsymbol{P}^{(i)\top} \boldsymbol{P}^{(i)})}_{=\boldsymbol{I}} \boldsymbol{x}_{vd}^{(i)}\|_2 \tag{26}$$

$$= \|\boldsymbol{P}^{(i)} \boldsymbol{U}_{v\bullet}^{(i)} w_{vj\bullet}(\boldsymbol{\Lambda}_{v\bullet}^{(i)}) \boldsymbol{U}_{v\bullet}^{(i)} \boldsymbol{x}_{vd}^{(i)}\|_2 \tag{27}$$

$$= \|\boldsymbol{U}_{v\bullet}^{(i)} w_{vj\bullet}(\boldsymbol{\Lambda}_{v\bullet}^{(i)}) \boldsymbol{U}_{v\bullet}^{(i)} \boldsymbol{x}_{vd}^{(i)}\|_2 \tag{28}$$

$$= \left(\boldsymbol{v}_\bullet^{(i)}\right)_{dj}, \tag{29}$$

where we used that $\boldsymbol{P}^{(i)}$ is norm-preserving to go from (27) to (28). Hence, the vertex spectral features $\boldsymbol{v}_\bullet^{(i)}$ are invariant under graph isomorphism and by the same argument, one can show that the edge spectral features $\boldsymbol{e}_\bullet^{(i)}$ are also invariant. Altogether, the Hodgelet spectral features (20) are well-defined features for graph inputs.

# D Extension to Higher-Order Networks

Graphs can support signals on their vertices and edges, resulting in a richer structure than tabular data. Common examples are social networks, electrical grids, citation networks, traffic maps, chemical reaction networks, collaboration networks, and molecules. However, graphs do have an important limitation: a graph only permits *dyadic interactions* between its vertices via its edges. Multiple recent works have tackle this issue by focusing on more general structures, such as *simplicial complexes*, *cellular complexes* and *hypergraphs* [30, 22, 18, 19, 16, 15, 24, 23, 31]. For a comparative illustration between a graph, simplicial complex and a cellular complex, we refer to Figure D.1.

## D.1 Simplicial complexes

Simplicial complexes represent a natural extension of graphs, and generalise the notion of vertices and edges by using *simplices* or *simplex* in the singular. In this broader setting, vertices are referred to as 0-simplices, and edges as 1-simplices. A 2-simplex is a triangle formed by three adjacent vertices. We observe that this describes a *triadic interaction* between vertices, in other words, a relation between three vertices. We can create simplices of arbitrary dimension $k \geqslant 0$. However,

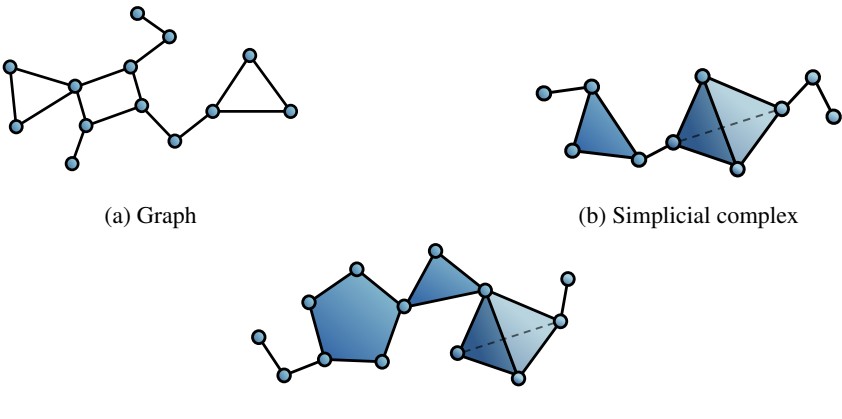

(a) Graph

(b) Simplicial complex

(c) Cellular complex

Figure D.1: Graph, simplicial complex, and cellular complex (specifically, a polyhedral complex). A simplicial complex cannot represent arbitrary polygons like the pentagon in (c).

the first three dimensions are often sufficient. We define a simplicial complex $\mathcal{S} = (\mathcal{V}, \mathcal{E}, \mathcal{T})$ has a structure constructed from a vertex set $\mathcal{V}$, an edge set $\mathcal{E}$, and a triangle set $\mathcal{T}$. We say that $\mathcal{S}$ is a simplicial 2-complex or a simplicial complex of dimension 2. Finally, the number of vertices, edges, and triangles are denoted by $N_v, N_e$ and $N_t$, respectively.

**Orientation.** So far, our simplicial complexes are *undirected*. However, we can give an *orientation* to a simplicial complex (see Figure D.2). An oriented simplicial 2-complex has an orientation on its edges and triangles, requiring an ordering to be imposed on its vertices. For an edge $e = \{v_1, v_2\}$, an orientation means assigning a direction, transforming $e$ into either $(v_1, v_2)$ or $(v_2, v_1)$. We notice that an edge always has exactly two possible orientations. A triangle $t = \{v_1, v_2, v_3\}$ similarly also has two possible orientations, corresponding to the parity of permutations of $\{v_1, v_2, v_3\}$. The parity defines two equivalence classes

$$[(v_1, v_2, v_3)] = \{(v_1, v_2, v_3), (v_2, v_3, v_1), (v_3, v_1, v_2)\}, \tag{30}$$

$$[(v_3, v_2, v_1)] = \{(v_3, v_2, v_1), (v_2, v_1, v_3), (v_1, v_3, v_2)\}, \tag{31}$$

which, at an intuitive level, corresponds to orientations that are either *clockwise* or *anti-clockwise*. We point out that although an orientation is necessary to express concepts such as edge flows, the choice of orientation is arbitrary. Fixing an orientation is akin to converting an undirected graph into a *directed* one, although they are fundamentally different since oriented graphs do not permit two edges between the same pair of vertices, whereas directed graphs do.

**Cellular complexes.** A more general structure is that of cellular complexes. The treatment of cellular complexes is similar to simplicial complexes. For more details, see Alain et al. [16]. We could also easily handle hypergraphs by using a Laplacian on hypergraphs.

### D.2 Hodge Laplacians

If one takes a step back, it becomes apparent that graphs and simplicial complexes have a *chain-like* structure: vertices are connected to edges and edges are connected to triangles. More precisely, we say that vertices have edges as their *upper neighbours*, edges have vertices as their *lower neighbours* and triangles as their upper neighbours, and triangles have edges as their lower neighbours. This leads us to the definition of two incidence matrices, $\boldsymbol{B}_{ve} \in \mathbb{Z}^{N_v \times N_e}$ and $\boldsymbol{B}_{et} \in \mathbb{Z}^{N_e \times N_t}$, sometimes called vertex-to-edge and edge-to-triangle incidence matrices, respectively. While $\boldsymbol{B}_{ve}$ is the incidence matrix between vertices and edges, $\boldsymbol{B}_{et}$ is the incidence matrix between edges and triangles. In a more general setting, these matrices are simply called *boundary matrices* and they are denoted by $\boldsymbol{B}_1$ and $\boldsymbol{B}_2$, respectively, since vertices are 1-simplices and edges are 2-simplices. This results in the general definition of a boundary matrix $\boldsymbol{B}_k \in \mathbb{Z}^{N_k \times N_{k+1}}$, where $N_k$ is the number of $k$-simplices. Note that for a simplicial $K$-complex, $\boldsymbol{B}_0 = \boldsymbol{0}$ and $\boldsymbol{B}_{k+1} = \boldsymbol{0}$. This simply means that 0-simplices have no lower neighbours, and $k$-simplices have no upper neighbours. We can define a more general

notion of Laplacian, called the *Hodge Laplacian*,

$$L_k := \mathbf{B}_k^\top \mathbf{B}_k + \mathbf{B}_{k+1}\mathbf{B}_{k+1}^\top \in \mathbb{Z}^{N_k \times N_k}. \tag{32}$$

The component $\mathbf{B}_k^\top \mathbf{B}_k$ is sometimes called the *lower Laplacian*, and $\mathbf{B}_{k+1}\mathbf{B}_{k+1}^\top$, the *upper Laplacian*. We observe in particular that the graph Laplacian is nothing other than $L_0$, and the graph Helmholtzian is $L_1$.

Interested readers are invited to find out more by looking at the concept of *chain complex*.

**Cliques.** A $k$-clique is a subset of $k$ vertices, such that every $k$ distinct vertices in the clique are adjacent. Intuitively, 2-cliques are like considering triangles. Edges have no upper neighbours in simplicial 1-complexes, i.e. graphs. Hence, the edge-to-triangle incidence matrix $\mathbf{B}_2 = \mathbf{0}$. In turn, this implies that the graph Helmholtzian is equal to its lower Laplacian, i.e. $L_1 = \mathbf{B}_1^\top \mathbf{B}_1$. Nevertheless, we can consider the triangles of a graph in order to have a non-zero $\mathbf{B}_2$ (see Figure D.2). We can then have a complete graph Helmholtzian, i.e. $L_1 = \mathbf{B}_1^\top \mathbf{B}_1 + \mathbf{B}_2\mathbf{B}_2^\top$.

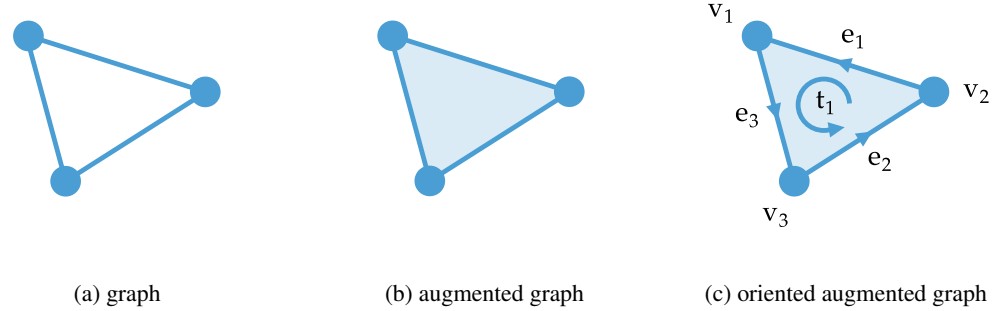

(a) graph  (b) augmented graph  (c) oriented augmented graph

Figure D.2: A graph is augmented by taking its 2-cliques and then assigned an orientation.

### D.3 Hodge decomposition

The *Helmholtz decomposition* is often called the fundamental theorem of vector calculus. This famous theorem states that any 2D vector fields on a Euclidean domain can be expressed as the sum of two orthogonal components: (1) one that intuitively has no sinks or sources, called *divergence-free* or *solenoidal*, and (2) one that has no vortices, sometimes called *curl-free* or *irrotational*. This decomposition extends to differential forms on manifolds and is referred to as the *Hodge decomposition*. For Riemannian manifolds, we can identify differential 1-forms with vector fields via the musical isomorphism. This implies that Hodge decomposition can be interpreted as a decomposition of vector fields on Riemannian manifolds. The Hodge decomposition on the $k$-simplex feature space $\mathbb{R}^{N_k}$ (for a single channel) leads to the sum of three orthogonal components: *exact*, *co-exact*, and *harmonic*,

$$\mathbb{R}^{N_k} = \mathrm{im}(\boldsymbol{B}_{k+1}) \oplus \mathrm{im}(\boldsymbol{B}_k^\top) \oplus \ker(\boldsymbol{L}_k), \tag{33}$$

where $\mathrm{im}(\boldsymbol{B}_{k+1})$ is the *exact subspace*, $\mathrm{im}(\boldsymbol{B}_k^\top)$ is the *co-exact subspace*, and $\ker(\boldsymbol{L}_k)$ is the *harmonic subspace*. Interestingly, we note that the dimension of $\ker(\boldsymbol{L}_k)$ is equal to the $k$-th *Betti number*, which describe the number of $k$-dimensional holes. The Hodge decomposition implies the eigendecomposition of the Hodge Laplacian,

$$L_k = \begin{pmatrix} \boldsymbol{U}_{ke} \\ \boldsymbol{U}_{kc} \\ \boldsymbol{U}_{kh} \end{pmatrix} \begin{pmatrix} \boldsymbol{\Lambda}_{ke} & \mathbf{0} & \mathbf{0} \\ \mathbf{0} & \boldsymbol{\Lambda}_{kc} & \mathbf{0} \\ \mathbf{0} & \mathbf{0} & \boldsymbol{\Lambda}_{kh} \end{pmatrix} \begin{pmatrix} \boldsymbol{U}_{ke} \\ \boldsymbol{U}_{kc} \\ \boldsymbol{U}_{kh} \end{pmatrix}^\top, \tag{34}$$

where $(\boldsymbol{\Lambda}_{ke}, \boldsymbol{U}_{ke})$ are the non-zero eigenvalues and eigenvectors of $\boldsymbol{B}_k^\top \boldsymbol{B}_k$, $(\boldsymbol{\Lambda}_{kc}, \boldsymbol{U}_{kc})$ are the non-zero eigenvalues and eigenvectors of $\boldsymbol{B}_{k+1}\boldsymbol{B}_{k+1}^\top$, and $(\boldsymbol{\Lambda}_{kh}, \boldsymbol{U}_{kh})$ are the zero eigenvalues and eigenvectors of $\boldsymbol{L}_k$. We observe that the *exact eigenbasis* $\boldsymbol{U}_{0e} = \mathbf{0}$ as $\boldsymbol{B}_0 = \mathbf{0}$. Similarly, *co-exact eigenbasis* $\boldsymbol{U}_{k+1e} = \mathbf{0}$ since $\boldsymbol{B}_{k+1} = \mathbf{0}$. We also have that $\boldsymbol{U}_{ke}$ spans $\mathrm{im}(\boldsymbol{B}_{k+1})$, $\boldsymbol{U}_{kc}$ spans $\mathrm{im}(\boldsymbol{B}_k^\top)$, and $\boldsymbol{U}_{kh}$ spans $\ker(\boldsymbol{L}_k)$.

**Hodgelet kernel.**    In the same spirit as in Section 2.1, we compute the wavelet transform on the $k$-simplex features and then obtain the wavelet spectral features. Finally, we present the *Hodgelet kernel* on simplicial $K$-complexes,

$$\kappa\big(\mathcal{S}^{(i)}, \mathcal{S}^{(j)}\big) \coloneqq \sum_{k=1}^{K} \Big( \kappa_{ke}\big(\boldsymbol{s}_{ke}^{(i)}, \boldsymbol{s}_{ke}^{(j)}\big) + \kappa_{kc}\big(\boldsymbol{s}_{kc}^{(i)}, \boldsymbol{s}_{kc}^{(j)}\big) + \kappa_{kh}\big(\boldsymbol{s}_{kh}^{(i)}, \boldsymbol{s}_{kh}^{(j)}\big) \Big), \tag{35}$$

where $\boldsymbol{s}_{k\bullet}^{(i)}$ for $\bullet \in \{e, c, h\}$ are spectral features corresponding to the $k$-simplices. This is defined in an analogous way to the construction detailed in Appendix C.

# E    Experiment Details

## E.1    Graph Classification Benchmarks

We refer to Opolka et al. [5] for details on all the datasets used in this section.

## E.2    Vector Field Classification Details

We provide some details about the vector field classification experiment in Section 3. To generate a random vector field, we first sample a Gaussian process $f : \Omega \times \mathbb{R}^2 \to \mathbb{R}^2$, and then take its derivatives $f_x, f_y$ in both spatial components. Noting that a curl-free 2D vector field is always the gradient of a potential, we can sample an arbitrary *curl-free field* by taking

$$\boldsymbol{X}_{\text{curl-free}} \coloneqq \nabla f = [f_x, f_y]^\top. \tag{36}$$

Similarly, since a *divergence-free* 2D vector field is always a *Hamiltonian vector field*, we can choose

$$\boldsymbol{X}_{\text{div-free}} \coloneqq \nabla^\perp f = [f_y, -f_x]^\top, \tag{37}$$

in order to sample an arbitrary divergence-free field.

For the vector field $f$, we used samples of the squared-exponential Gaussian process, sampled using its random feature approximation [32]. We display this in Figure E.1, where we plot the derivatives $f_x, f_y$, and their combinations to yield divergence-free and curl-free fields.

Our data consist of *mostly divergence-free and curl-free* vector fields, which are generated by considering the linear combination

$$\boldsymbol{X} \coloneqq \lambda \boldsymbol{X}_{\text{div-free}} + (1 - \lambda)\boldsymbol{X}_{\text{curl-free}} + R\epsilon, \tag{38}$$

where $\lambda \sim U([0.1, 0.9])$, $R > 0$ is the noise level, $\epsilon \sim \mathcal{N}(0, 1)$, and $\boldsymbol{X}_{\text{div-free}}, \boldsymbol{X}_{\text{curl-free}}$ generated randomly. If $\lambda < 0.5$, we say that the vector field $\boldsymbol{X}$ is mostly curl-free, and if $\lambda > 0.5$, we say that it is mostly divergence-free.

**Projecting to a simplicial complex.**    To project a vector field $\boldsymbol{X} : \mathbb{R}^2 \to \mathbb{R}$ onto the edges of a graph, we employ the *de Rham map* [33], which "discretises" a vector field into edge signals. For a given oriented edge $e = (v_0, v_1) \in \mathcal{E}$ with endpoint coordinates $\boldsymbol{x}_0$ and $\boldsymbol{x}_1$, we define the projection $X^e$ of $\boldsymbol{X}$ onto $e$ by the following integral,

$$X^e \coloneqq \int_0^1 \boldsymbol{X}\big(\boldsymbol{x}_0 + t(\boldsymbol{x}_1 - \boldsymbol{x}_0)\big) \cdot \hat{\boldsymbol{t}} \, \mathrm{d}t, \tag{39}$$

where $\hat{\boldsymbol{t}} \coloneqq \frac{\boldsymbol{x}_1 - \boldsymbol{x}_0}{\|\boldsymbol{x}_1 - \boldsymbol{x}_0\|}$ is the unit tangent vector along the edge. Numerically, this can be computed efficiently using numerical quadratures, owing to the fact that the integral is only defined over the interval $[0, 1]$. Note that this depends on the ordering of the vertices $v_0$ and $v_1$ characterising the edge – if we flip the order, then the sign of $X^e$ flips. Thus, we require the graph to be oriented in order for the projections $\{X^e\}_{e \in \mathcal{E}}$ to be well-defined. In Figure E.2, we display an example of such a projection onto a regular triangular mesh.

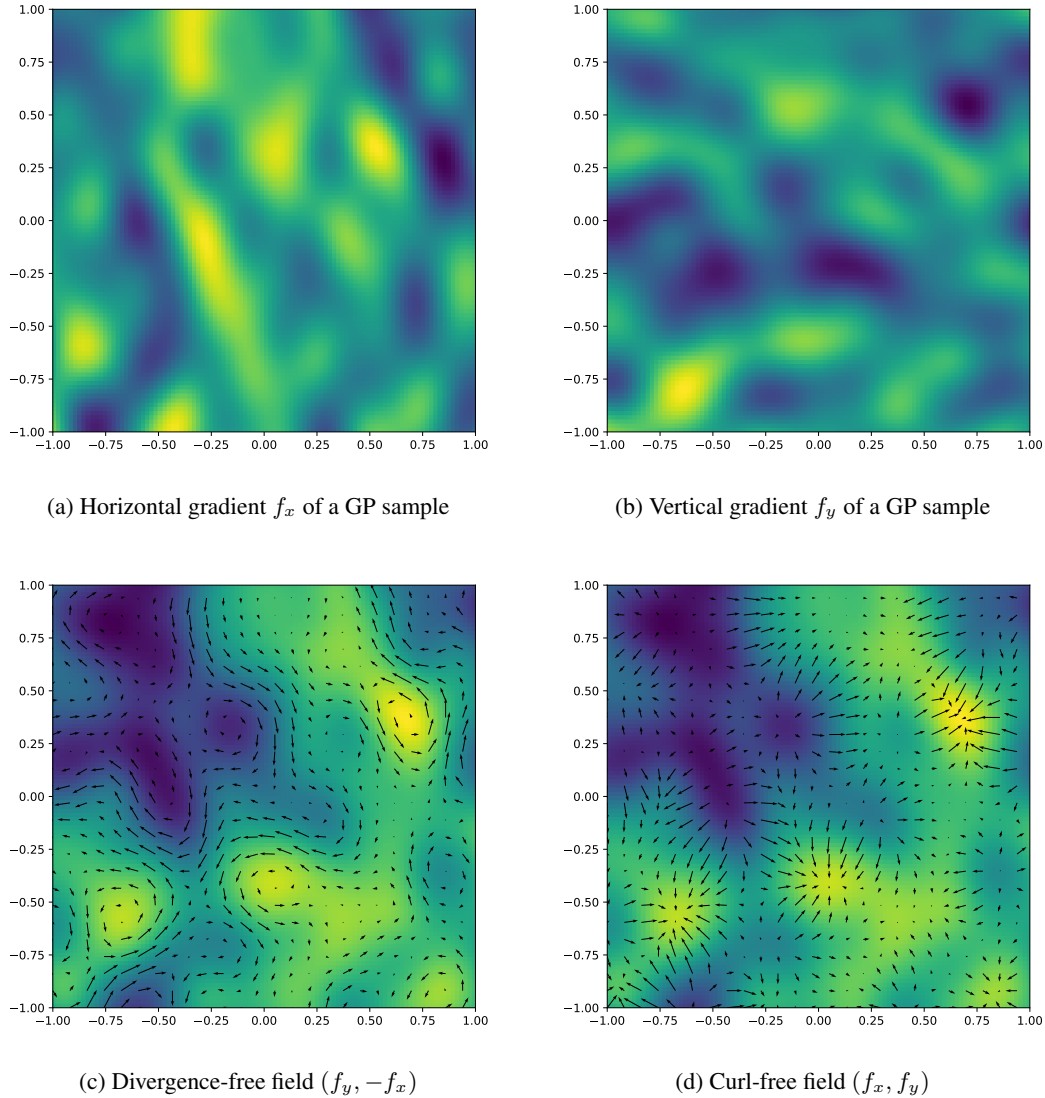

(a) Horizontal gradient $f_x$ of a GP sample

(b) Vertical gradient $f_y$ of a GP sample

(c) Divergence-free field $(f_y, -f_x)$

(d) Curl-free field $(f_x, f_y)$

Figure E.1: Illustration of the random vector field data generating process.

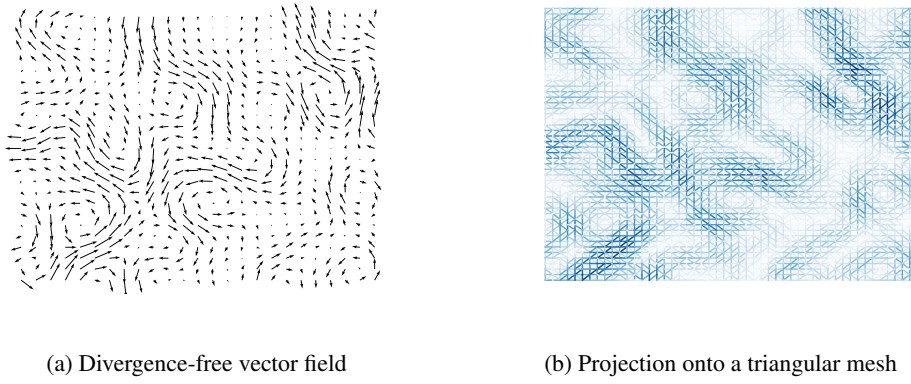

(a) Divergence-free vector field

(b) Projection onto a triangular mesh

Figure E.2: Projection of a continuous divergence-free field onto a regular triangular mesh.

