# OpenReview forum: "Graph Classification Gaussian Processes via Hodgelet Spectral Features"
_NeurIPS.cc/2024/Workshop/BDU — NeurIPS BDU Workshop 2024 Oral_

### Official Review · Reviewer_ETym · 2024-09-19

**Rating:** 7
**Confidence:** 3

**Review:**

The paper proposes a Hodge decomposition graph classification algorithm within a Gaussian process framework that captures signals on vertices, edges, and higher-order graph structures by analyzing their projections onto the eigenbasis of the graph Laplacian and Helmholtzian. The key idea is to convert the non-Euclidean graph signals into the ones in the Euclidean domain and then apply the standard GP classification algorithm. The paper introduces graph wavelet transforms of the vertex and edge signals and further generates features based on Hodge decomposition as an input to downstream tasks. Experimental results validate the effectiveness of the WT-GP-Hodge algorithm as well as its improvement over the existing methods.

**Strengths**

1. The algorithm directly processes vertex and edge signals, as opposed to WT-GP, which needs to transform the original graph into a line graph. The algorithmic flexibility and practicality are enhanced.
2. The idea of Hodge decomposition and graph wavelet transforms before a standard GP is simple and interesting.
3. The paper is well-written and the results are presented clearly.

**Weaknesses**

1. The method can tackle vertex and edge signals. However, only an algorithm incapable of processing edge features is compared. Empirically, it might be better to conduct an experiment to illustrate that WT-GP can also beat an algorithm incapable of processing vertex features.

2. The authors claim that WT-GP can incorporate high-order graph features and briefly discuss the methodology in Appendix B, which essentially replaces Laplacian / Helmholtzian with the k-th Hodge Laplacian. I am a little skeptical about the kernel in (12) losing high-order information. Experimental results should be added to illustrate this point.

---

### Official Review · Reviewer_1d8p · 2024-09-26
**Review of GP Graph Classification with Hodge Decomposition on vertex and edge spectral signals**

**Rating:** 9
**Confidence:** 5

**Review:**

This paper presents a novel approach for graph classification with GP with spectral features constructed from both vertex and edges of the graph. It extends the interesting approach of the paper Graph classification gaussian processes via spectral features. It not only extends the features to include edges in addition to vertex, but also introduces the Hodge decomposition approach and formulates the GP classification model with additive kernels. Several interesting approaches and novel ideas are proposed and integrated by this paper to improve further on GP graph classification.

The paper also compares its approach and previous approach in several benchmark dataset extensively to demonstrate the classification improvement. In addition, this paper also designs the simulation of vector fields as evaluation dataset of graph classification. It shows the performance improvement on the vector field classification. Further discussions on higher-order networks are also presented.

---

### Decision · Program_Chairs · 2024-10-09

Accept (Oral)